# SHARED EMBEDDING OPTIMIZATION: A TWO-STAGE APPROACH FOR EFFICIENT AND EFFECTIVE FEATURE EMBEDDING

## ABSTRACT

Large-scale recommendation systems are dominated by the memory and computational cost of their embedding tables. Standard approaches force a difficult compromise: either use independent embedding tables for each feature ("Separate"), which offers high performance but can be inefficient in terms of parameters, or use a single table shared by all features ("Monolithic Shared"), which is parameter-efficient but imposes a naive, homogeneous structure that degrades model quality. This paper argues that this is a false dilemma. We contend that the structure of a shared embedding table should be neither fully separate nor fully monolithic, but rather a learned, heterogeneous configuration optimized for the task. We propose *Shared Embedding Optimization (SEO)*, a novel two-stage framework to discover and instantiate such a structure automatically. In Stage 1 (Search), we train an over-parameterized model to learn an optimal, feature-specific policy for sharing and allocating embedding chunks. In Stage 2 (Applier), we instantiate a new, compact model based on this learned policy and retrain it from scratch. We provide a theoretical justification showing our Applier's hypothesis space is a superset of the monolithic model. Comprehensive experiments on three large-scale benchmarks (Criteo, Avazu, MovieLens 1M) demonstrate that SEO significantly and consistently outperforms both 'Separate' and 'Monolithic Shared' baselines, given an identical embedding parameter budget and a comparable recurring training time.

## 1 INTRODUCTION

Embedding layers are the cornerstone of modern deep learning models for categorical data, particularly in large-scale systems like click-through rate (CTR) prediction (Cheng et al., 2016; Anil et al., 2022). These embedding tables map high-cardinality sparse features (e.g., user_id, item_id) to dense vectors and often consume over 99% of total model parameters. At industrial scale, they can grow to trillions of parameters, consuming petabytes of memory (Mudigere et al., 2022). This massive size creates a bottleneck for model training, inference, and deployment, making parameter efficiency a first-order research priority.

Current approaches to managing this parameter budget typically force practitioners into an unfortunate compromise between two opposing philosophies. On one hand, **Separate Embeddings** assign an independent embedding table to each categorical feature. While this allows each feature to learn a specialized representation and often yields high performance, it is immensely parameter-inefficient, fails to exploit potential inter-feature similarities, and scales poorly as feature count and cardinality grow. On the other hand, **Monolithic Shared Embeddings (MSE)** utilize a single, unified embedding table for all features—a strategy refined and successfully deployed at web-scale (Coleman et al., 2023). This approach is highly parameter-efficient but imposes a naive, 'one-size-fits-all' structure. It forces features with vastly different characteristics—such as a high-cardinality user_id and a low-cardinality day_of_week—to share the same parameter space and capacity, leading to representation collisions and a demonstrable degradation in model quality.

This paper argues that the choice between full separation and monolithic sharing represents a **false dilemma**. We contend that the optimal solution lies not at these extremes, but in a *heterogeneous* embedding structure, where the parameter budget is intelligently allocated and shared based on the

specific needs of each feature and the overall task. To this end, we propose **Shared Embedding Optimization (SEO)**, a novel two-stage framework that breaks this dilemma by *learning* this optimal, heterogeneous structure automatically while adhering to a fixed total parameter budget.

The SEO framework operates in two stages.

- **Stage 1 (Search):** We design an over-parameterized search space where a base embedding table is partitioned into many small 'chunks'. We then employ a differentiable search mechanism, adapted from the structured pruning literature (Yasuda et al., 2023; 2024), to learn a sparse, feature-specific attention policy that discovers the optimal allocation and sharing pattern for each feature.

- **Stage 2 (Applier):** After the search, we discard the search model's weights entirely. Instead, we use the learned policy as a blueprint to instantiate a *new, compact Applier model* that contains only the embedding chunks selected by the policy. This new, structurally-efficient model is then **retrained from scratch** to converge to a superior solution.

Our contributions are as follows:

1. We propose SEO, a novel two-stage framework that learns a task-specific, heterogeneous embedding structure to optimize the performance-per-parameter trade-off.

2. We provide a theoretical justification (Section 3.3) showing that our Applier model's hypothesis space is a strict superset of the Monolithic Shared Embedding baseline, guaranteeing a solution of at least equal quality.

3. We conduct comprehensive experiments on three large-scale public benchmarks (Criteo, Avazu, MovieLens 1M). Our results (Section 4) show that, given an **identical parameter budget**, our Applier models significantly outperform both `Separate` and `Monolithic Shared` baselines (Table 1).

4. We present a detailed ablation study (Table 2) that validates our design, proving that the two-stage (Search then Retrain) process is essential and that the search mechanism is robust. It also reveals the distinct advantages of our two Applier variants, `Applier (RR)` and `Applier (Exact)`, on different types of datasets.

5. We demonstrate (Table 3) that our method's superior performance does not come at a high cost; the **recurring training time** of our final Applier model is competitive with (and in some cases, faster than) standard baselines.

## 2 RELATED WORK

The immense scale of embedding tables in deep learning recommender systems (DLRS) is a central challenge, with models reaching trillions of parameters (Mudigere et al., 2022). Research to mitigate this has broadly followed two paths: reducing parameters via weight sharing and automating the design of the embedding architecture itself. A foundational approach to parameter reduction is the hashing trick (Moody & Darken, 1989; Weinberger et al., 2009), which maps features to a fixed-size table but suffers from random collisions. This monolithic sharing philosophy has been refined and successfully deployed in industrial-scale systems like Unified Embedding (Coleman et al., 2023), which use a single, shared representation space. While highly effective, this "one-size-fits-all" structure forces features with diverse characteristics to share capacity, creating an information bottleneck. To address this, a more advanced paradigm uses automated machine learning (AutoML) to discover an optimal embedding architecture, inspired by Neural Architecture Search (NAS) (Zoph & Le, 2017; Liu et al., 2019). The most closely related works in this area are frameworks like AutoEmb (Zhao et al., 2021b) and AutoDim (Zhao et al., 2021a), which automatically assign a variable embedding dimension to each feature. These methods, however, solve a different problem: they ask, "What is the optimal embedding *dimension (width)* for each feature's *separate* table?" This typically results in multiple tables of varying widths.

In contrast, our work, Shared Embedding Optimization (SEO), addresses a distinct and complementary question:

> Given a *single, unified embedding table* with a *fixed total parameter budget*, what is the optimal policy for allocating and sharing its *internal subspaces* (chunks) among features?

This focus on learning the internal structure of a shared representation space under a strict budget constraint is the core differentiator of our work. Our work is also inspired by recent advances in differentiable structured pruning, particularly Yasuda et al. (2024), from which we adapt the core differentiable search mechanism. However, it is crucial to distinguish our contribution. Methods like SequentialAttention++ aim to prune an existing, fixed-size model by identifying and removing unimportant weight blocks to improve inference efficiency. In contrast, SEO addresses the distinct problem of embedding architecture search. We do not prune a model in-place; instead, we leverage the differentiable search to discover an optimal architectural blueprint for a shared embedding table. This blueprint is then used to instantiate and train a new, compact, and structurally superior model from scratch. Our contribution is thus the novel two-stage search-and-retrain framework for embedding tables, not the search mechanism itself. For a more detailed discussion of foundational compression techniques and other structured sharing methods, please see Appendix B.

## 3 Method: Shared Embedding Optimization (SEO)

Shared Embedding Optimization (SEO) is a two-stage framework designed to discover and instantiate a superior, heterogeneous structure for a unified embedding table, while strictly adhering to a predefined parameter budget. The core principle is to decouple the complex problem of architectural discovery from the final task of weight optimization. First, in a search stage, we explore an over-parameterized space to identify an optimal, feature-specific policy for sharing and allocating embedding subspaces. Second, using this policy as a blueprint, we construct a new, compact, and structurally efficient model, which is then retrained from scratch to achieve superior performance. Figure 1 provides a conceptual overview of baseline methods and the key components of our SEO framework.

### 3.1 Stage 1: Search for an Optimal Sharing Policy

**Search Space Definition.** We start by defining the total parameter budget, $P = B_{base} \times D$, which equals that of a standard **Monolithic Shared Embedding (MSE)** model with an embedding table $W_H \in \mathbb{R}^{B_{base} \times D}$. To create a sufficiently rich search space for discovering novel structures, we construct an over-parameterized **search model**, $M_O$. This is achieved by expanding the embedding dimension from $D$ to a larger search dimension, $D_{search}$, typically a multiple of $D$. This expanded table, $W_O \in \mathbb{R}^{B'_{base} \times D_{search}}$, provides the necessary representation capacity for the search.

We then logically partition each row of this widened table into $N'$ non-overlapping vectors, which we term **chunks**. These $N'$ partitions form a **shared pool of candidate chunk types**, each with dimension $d_{chunk}$ (where $N' \times d_{chunk} = D_{search}$). The objective of the search stage is to learn a distinct selection policy for each feature over this common pool of chunks.

**Differentiable Structure Search.** To learn a good chunk allocation for each feature, we employ a differentiable search mechanism adapted from the structured pruning literature (Yasuda et al., 2023; 2024). We introduce a matrix of learnable logits, $\mathbf{L} \in \mathbb{R}^{F \times N'}$, where $F$ is the number of categorical features. The importance score, or attention weight $\alpha_{f,i}$ for the $i$-th chunk of the $f$-th feature, is computed via a global softmax across all possible chunks for all features:

$$\alpha_{f,i} = \frac{\exp(L_{f,i}/\tau)}{\sum_{f'=1}^{F} \sum_{j=1}^{N'} \exp(L_{f',j}/\tau)} \tag{1}$$

where $\tau$ is a temperature parameter. This global normalization ensures that all $F \times N'$ weights form a single probability distribution. Yasuda et al. (2024) show that this is equivalent to imposing a sparsity-inducing log-sum regularizer, which encourages the attention distribution $\alpha$ to become sparse during training. This allows the model to identify and prioritize the most salient chunks for each feature.

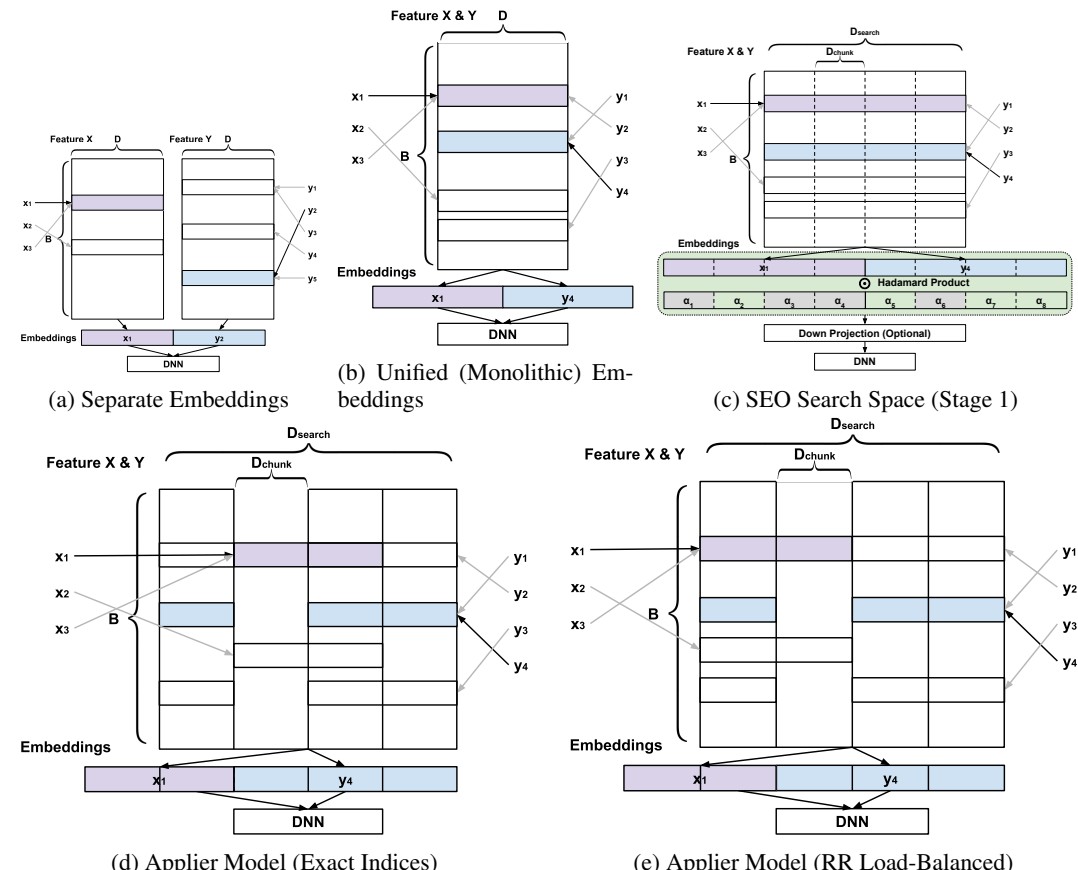

(a) Separate Embeddings

(b) Unified (Monolithic) Embeddings

(c) SEO Search Space (Stage 1)

(d) Applier Model (Exact Indices)

(e) Applier Model (RR Load-Balanced)

Figure 1: **Illustration of different embedding table structures.** The diagrams show lookups for different input values (e.g., $x_1, x_2$ are values for Feature X; $y_1, y_4$ are values for Feature Y). (a) The **Separate Embeddings** baseline assigns an independent table to each feature. (b) The **Monolithic Shared** baseline uses a single table for all features and their values. (c) Our **SEO Stage 1 (Search)** operates in an over-parameterized space ($D_{search}$), partitioned into multiple "chunks." It learns a policy to select the optimal chunks for each feature's values. (d) and (e) depict the compact **SEO Stage 2 (Applier)** models built from this policy. (d) The **Applier (Exact Indices)** variant instantiates the precise chunk assignments discovered during search (e.g., values for Feature X use chunks 1, 2, and 4, while values for Feature Y use chunks 1 and 3). (e) The **Applier (RR Load-Balanced)** variant uses the learned *number* of chunks per feature but reassigns them in a round-robin manner to balance table utilization.

During the forward pass, the final embedding for an input $x_f$ of feature $f$ is constructed by a weighted concatenation of its looked-up chunks $\{C_1(x_f), \ldots, C_{N'}(x_f)\}$:

$$E_f(x_f) = \text{Concat}\left(\alpha_{f,1}C_1(x_f), \ldots, \alpha_{f,N'}C_{N'}(x_f)\right) \tag{2}$$

The resulting vector $E_f(x_f)$ has dimension $D_{search}$. These feature-specific embeddings are then passed to the downstream network.

**Policy Derivation.** Once the search model has converged, the learned logits $\mathbf{L}$ represent a continuous distribution of chunk importance. We derive a discrete architectural policy from these scores. For each feature $f$, we identify the set of chunk indices with the highest corresponding attention scores $\{\alpha_{f,i}\}$. This final, discrete policy serves as the blueprint for constructing the new model in Stage 2.

**Practical Considerations for the Search Phase.** Two configurations are introduced to manage computational constraints and ensure experimental integrity during the search.

- **Use Down Projection (Memory-Efficient Search):** The search model's expanded embedding width can cause out-of-memory errors when feature embeddings are concatenated. To mitigate this, we provide the use_down_projection option. Instead of viewing the output as $F$ wide embeddings, we consider the complete pool of $F \times N'$ candidate chunks. For a given example, the set of looked-up vectors can be represented as a matrix $\mathbf{C}_{pool} \in \mathbb{R}^{(F \times N') \times d_{chunk}}$. We then apply a learnable linear projection matrix $\mathbf{P} \in \mathbb{R}^{(F \times N') \times M}$ to reduce this to a smaller set of $M$ vectors, where $M$ is the target number of chunks for the final model. The resulting projected vectors, $\mathbf{C}_{proj} \in \mathbb{R}^{M \times d_{chunk}}$, are calculated as:

$$\mathbf{C}_{proj} = \mathbf{P}^T \mathbf{C}_{pool}$$

  These $M$ vectors are then fed to subsequent layers, significantly reducing the memory footprint. This projection is a temporary scaffold used only during the search.

- **Adjust Hash Buckets (Optional for Search):** In our experiments, the search model's embedding table is widened to provide candidate chunks. To maintain a comparable parameter count against baselines during the search phase, we can optionally adjust the number of hash buckets downwards to compensate. This is an optional setting for the search model, as its primary purpose is exploration, not direct performance comparison.

## 3.2 STAGE 2: APPLIER MODEL INSTANTIATION AND RETRAINING

In the second stage, we discard the entire search model ($M_O$), including its embedding table and learned logits. We use the discrete policy derived in Stage 1 to construct a new, compact **Applier model**, $M_A$, from scratch.

**Satisfying the Parameter Budget.** A core tenet of our method is to ensure a fair comparison. The Applier model is meticulously constructed to have the **same parameter budget**, $P = B_{base} \times D$, as the baselines. To achieve this, it is **mandatory** to apply the adjust_hash_buckets strategy. If the learned policy results in a model with a total embedding width different from $D$, we adjust the number of hash buckets in the Applier's embedding table to precisely compensate, thereby ensuring the total parameter count remains strictly $P$. This guarantees a direct and fair comparison against the baseline models.

**Assignment Strategies and Retraining from Scratch.** We explore two strategies for instantiating the learned policy within the Applier model, which has $M$ available embedding sub-tables (slots):

- **Applier (Exact Indices)**: This strategy faithfully preserves the specific chunk indices discovered during the search. If feature $F_1$ selected slots $\{2, 3\}$ in the search space, its embedding in the Applier model is constructed exclusively from the corresponding slots $T_2$ and $T_3$, precisely realizing the learned structure.

- **Applier (RR Load-Balanced)**: This strategy abstracts the policy, retaining only the *number* of chunks selected per feature. It then reassigns chunks using a round-robin schedule. For instance, if $F_1$ requires 2 chunks and $F_2$ requires 3, $F_1$ is assigned slots $\{T_1, T_2\}$ and $F_2$ is assigned $\{T_3, T_4, T_1\}$ (wrapping around from a total of $M = 4$ slots). This respects the learned per-feature capacity while balancing the utilization of embedding slots.

Once the structure is fixed using one of these strategies, the final Applier model's weights are randomly initialized and it is **trained from scratch**. This retraining step is critical, as it allows the model to find a superior solution within the optimized architecture, unencumbered by the compromises made during the search phase.

## 3.3 THEORETICAL JUSTIFICATION

Our framework is theoretically grounded. The standard monolithic model can be viewed as a constrained, homogeneous special case of the heterogeneous models our Applier can represent.

**Definition 1.** *Let the total embedding parameter budget be $P = B_{base} \times D$, partitioned into $M$ chunks, $T_1, ..., T_M$, each of size $B_{base} \times d_{chunk}$ where $M \times d_{chunk} = D$.*

- *Let $\mathcal{M}_H$ be the **Monolithic Shared Embedding (MSE) hypothesis space**, where the embedding function for every feature $f \in \mathcal{F}$ is fixed to use all chunks: $E_f(\mathbf{x}) = Concat(T_1(\mathbf{x}_f), ..., T_M(\mathbf{x}_f))$.*

- *Let $\mathcal{M}_A$ be the **Applier hypothesis space**, where the embedding function for each feature $f$ is determined by a learnable policy $\mathcal{S}_f \subseteq \{1, ..., M\}$: $E_f(\mathbf{x}, \mathcal{S}_f) = Concat(\{T_i(\mathbf{x}_f) \mid i \in \mathcal{S}_f\})$.*

**Theorem 1.** *The Monolithic Shared Embedding (MSE) hypothesis space $\mathcal{M}_H$ is a strict subset of the Applier hypothesis space $\mathcal{M}_A$. That is, $\mathcal{M}_H \subset \mathcal{M}_A$. (The proof is deferred to Appendix C.)*

**Corollary 1** (Optimization Guarantee). *Given the same parameter budget $P$, the optimal model found in the Applier space, $M_A^* = \arg\min_{M_A \in \mathcal{M}_A} \mathcal{L}(M_A)$, has a training loss less than or equal to the optimal model found in the Monolithic space, $M_H^* = \arg\min_{M_H \in \mathcal{M}_H} \mathcal{L}(M_H)$. Concretely,*

$$\mathcal{L}(M_A^*) \leq \mathcal{L}(M_H^*). \tag{3}$$

*(The proof follows directly from Theorem 1 and is provided in Appendix C.)*

The goal of SEO is therefore to find a non-trivial, heterogeneous policy $\mathcal{S}^*$ that allows $M_A^*$ to strictly outperform $M_H^*$ in practice.

## 4 EXPERIMENTS

### 4.1 EXPERIMENTAL SETUP

**Datasets and Preprocessing.** We evaluate on three public benchmarks: **Criteo** (online advertising, ∼45M examples, 26 categorical/13 continuous features (Criteo Labs, 2014)), **Avazu** (click-prediction, ∼36M examples, 22 categorical features), and **MovieLens 1M** (collaborative filtering, 6 categorical features, framed as binary prediction for ratings $\geq 3$ (Harper & Konstan, 2015)). We embed all features. Preprocessing and data splits follow Wang et al. (2021) for Criteo/MovieLens and Song et al. (2019) for Avazu.

**Model Architecture and Compared Methods.** All models use a standard DNN (MLP with ReLU activations over concatenated embeddings) with a sigmoid output for binary classification. We compare four embedding strategies: (1) **Separate** (independent hashed table per feature), (2) **Monolithic Shared Embedding (MSE)** (one shared table), and our proposed two-stage method consisting of the (3) **Shared Optimizer** (Stage 1 search) and (4) **Shared Applier** (Stage 2 retrain). We evaluate two Applier variants: **Applier (Exact Indices)** and **Applier (RR Load-Balanced)**.

**Training and Evaluation.** All models are trained using the Adam optimizer with a binary cross-entropy loss. We evaluate performance using Area Under the ROC Curve (AUC) and LogLoss. To ensure a fair comparison and robustness, we conduct a comprehensive hyperparameter search. The MLP hidden layers are selected from ['768,256,128', '256,128', '128,64', '64'] for Criteo, ['1024,512,256', '512,256,128', '256,128,64'] for Avazu, and ['256,128', '128,64', '64,32'] for MovieLens. The key hyperparameter, **Base Bucket Size** ($B_{base}$), which defines the total parameter budget, is varied across [2k, 20k, 100k, 200k, 400k] for Criteo, [2k, 20k, 200k] for Avazu, and [1k, 6k, 14k] for MovieLens. For the baseline and final Applier models, we use a base embedding dimension of $D = 16$, with a chunk size of $d_{chunk} = 8$. For the Stage 1 search model, we use an expanded search dimension of $D_{search} = 32$ to provide a rich pool of candidate chunks. We report the mean metrics over 5 runs with different random seeds.

### 4.2 RESULTS AND ANALYSIS

**Performance Comparison.** The full hyperparameter comparisons are available in Appendix D. Figure 2 highlights the key findings on Test AUC. Table 1 provides a quantitative summary for a representative configuration from each dataset. As shown in Table 1 and Figure 2, our proposed **SEO-Applier models achieve the highest Test AUC and lowest Test LogLoss on all three datasets** for the representative configurations. The full results in Appendix D confirm this trend across nearly all hyperparameter combinations. The Applier models (blue and green bars in the plots) consistently outperform the 'Separate' and 'Monolithic Shared' baselines (grey bars).

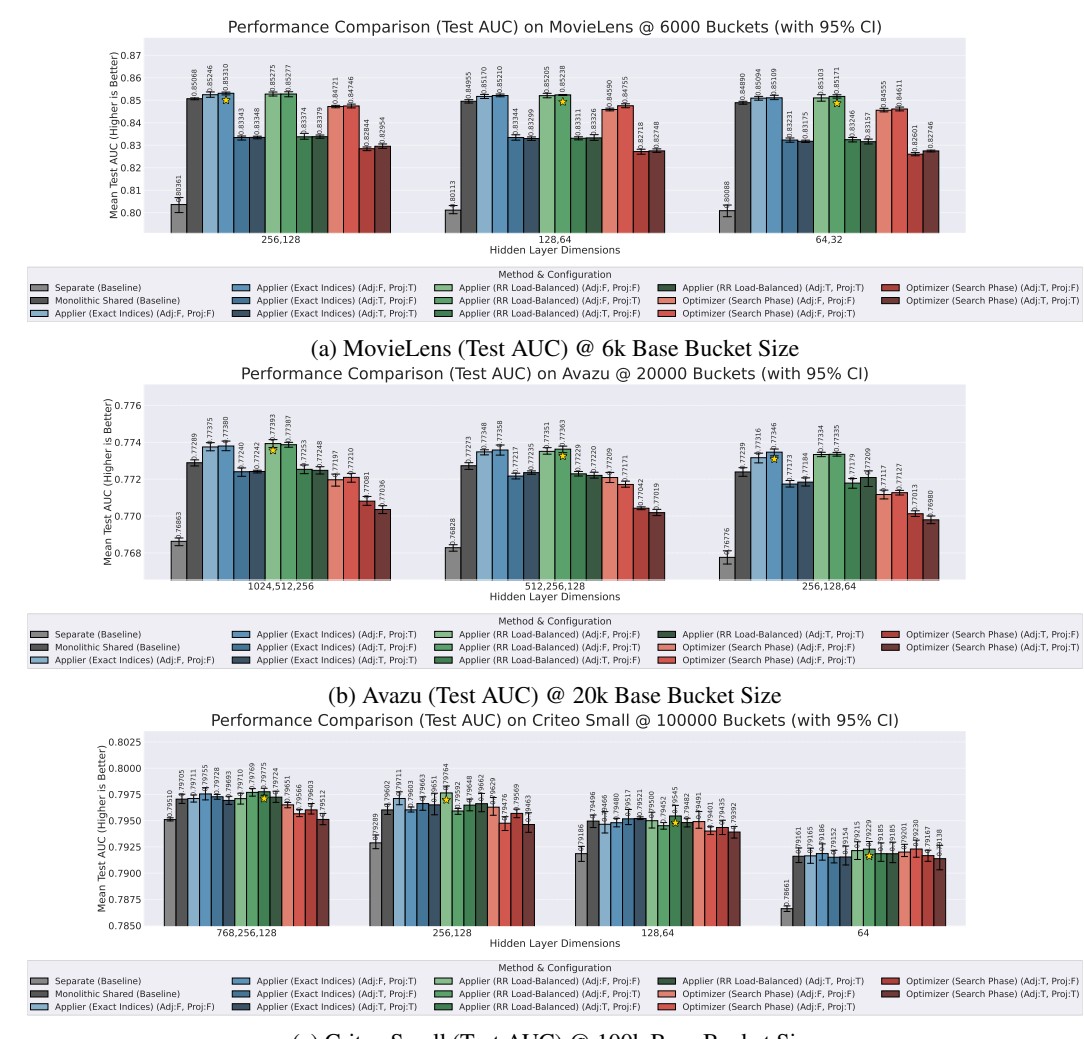

(a) MovieLens (Test AUC) @ 6k Base Bucket Size

(b) Avazu (Test AUC) @ 20k Base Bucket Size

(c) Criteo Small (Test AUC) @ 100k Base Bucket Size

Figure 2: Main results summary: Test AUC comparison on MovieLens, Avazu, and Criteo Small datasets for a representative base bucket size. Each plot shows performance across different hidden layer dimensions (x-axis) and methods (bars). In method names, "Adj" denotes adjust_hash_buckets and "Proj" denotes use_down_projection. The yellow stars highlight the best-performing Applier or Baseline model. Full hyperparameter comparisons are available in Appendix D.

**Embedding Parameter Budget.** We designed our experiments for a fair parameter comparison. All three main embedding strategies ('Separate', 'Monolithic Shared', and 'Applier') are allocated an **identical total embedding parameter budget**, defined as $P = B_{base} \times D$, where $B_{base}$ is the "base bucket size" (from the plots) and $D = 16$. For 'Monolithic Shared (MSE)' and our 'Applier' models, this budget is instantiated as a single unified table of size $B_{base} \times 16$. In contrast, for the 'Separate (Baseline)', this same total budget is distributed across the $F$ features; each feature is given an independent table of width $D = 16$ and a bucket size of $B_{feat} \approx B_{base}/F$, keeping the total parameter count identical ($F \times (B_{base}/F) \times 16 = B_{base} \times 16$). The results in Table 1 demonstrate our core finding: **given a fixed parameter budget**, our 'Applier' method significantly outperforms both the standard 'Separate' and 'Monolithic Shared' approaches in terms of Test AUC and LogLoss. The "efficiency" of our method lies in its superior *performance-per-parameter*, achieved by intelligently structuring the unified embedding table based on the search phase results.

**Ablation Study: Search Configuration and Applier Variants** To validate our two-stage design, we conduct a detailed ablation study across all three datasets, shown in Table 2. This study analyzes

| Dataset | Model | Test AUC | Test LogLoss | Params (M) |
|---|---|---|---|---|
| **Criteo** | Separate (Baseline) | 0.79510 | 0.45263 | 1.60 |
| (26 Feat, $B_{base}$=100k) | Monolithic Shared (MSE) | 0.79705 | 0.45119 | 1.60 |
| (H='768,256,128') | **Applier (RR Load-Balanced)** | **0.79775** | **0.45062** | 1.60 |
| **Avazu** | Separate (Baseline) | 0.76828 | 0.38359 | 0.32 |
| (22 Feat, $B_{base}$=20k) | Monolithic Shared (MSE) | 0.77273 | 0.38111 | 0.32 |
| (H='512,256,128') | **Applier (RR Load-Balanced)** | **0.77363** | **0.38063** | 0.32 |
| **MovieLens 1M** | Separate (Baseline) | 0.80361 | 0.41596 | 0.10 |
| (6 Feat, $B_{base}$=6k) | Monolithic Shared (MSE) | 0.85068 | 0.36787 | 0.10 |
| (H='256,128') | **Applier (Exact Indices)** | **0.85310** | **0.36590** | 0.10 |

Table 1: Performance comparison on representative configurations. We report the mean Test AUC and Test LogLoss over 5 random seeds. Best performance is in **bold**. All models within a dataset are allocated an identical embedding parameter budget.

two key axes: (1) The impact of the Stage 1 'Search' configuration (DP and AH) on the final result, and (2) the performance of our two proposed Stage 2 'Applier' variants. We observe three key findings:

- **Retraining (Stage 2) is Essential:** On all three datasets, all 'Applier' models (both 'Exact' and 'RR') consistently and significantly outperform their corresponding 'Search Model (Stage 1)'. For example, on Criteo with the '(False, True)' search configuration, the 'Search Model' achieves an AUC of 0.79603, while its 'Applier (RR)' counterpart improves to 0.79775. This confirms that the search model, designed for exploration, finds a suboptimal local minimum, and retraining from scratch is necessary to unlock the true performance of the discovered architecture.

- **Choice of Applier Variant Matters:** The 'Applier (RR Load-Balanced)' variant consistently achieves the best performance on the large-scale Criteo and Avazu datasets. Conversely, the 'Applier (Exact Indices)' variant performs best on the smaller MovieLens 1M dataset. This nuanced finding justifies our choice of using the optimal Applier variant for each dataset in Table 1. This behavior can be attributed to the characteristics of the datasets. Criteo and Avazu feature a large number of high-cardinality, sparse features, where the load-balancing of the RR strategy acts as a powerful regularizer against hash collisions. MovieLens, with fewer and denser features, benefits more from the precise structure discovered by the search, which the 'Exact' applier faithfully instantiates.

- **Search is Robust:** The choice of search parameters (DP/AH) has a minimal impact on the final 'Applier' model's performance. On all datasets, the 'Applier' results are extremely stable regardless of the search configuration (e.g., Avazu 'Applier (RR)' AUC is stable between 0.77220 and 0.77363 across all four search settings). This indicates that our search process is robust to these configuration choices.

Given these findings, we selected the best-performing Applier variant for each dataset (RR for Criteo/Avazu, Exact for MovieLens) as our primary model for all other experiments.

## 4.3 TRAINING OVERHEAD ANALYSIS

A key consideration for our two-stage framework is its computational cost. We divide this analysis into two parts: the recurring training cost of our final 'Applier' model (Stage 2), and the one-time, offline cost of the 'Search' phase (Stage 1). Our experiments demonstrate that our 'Applier' model achieves superior predictive performance with a recurring training time that is highly comparable to the baselines.

Table 3 details the per-epoch training times. To ensure robustness against systemic noise such as scheduling and pre-emption from running on a shared CPU cluster, we report the median time over 5 runs. The results show that the recurring training cost of our 'Applier' model is competitive. On Criteo and Avazu, the 'Applier' model's training time (1,738.3s and 2,122.1s, respectively) is slightly slower than the fastest baseline, MSE (**1,527.1s** and **1,949.2s**), but remains substantially faster than

| Dataset | Search Config (DP, AH) | Search Model | Applier Model (Stage 2) | |
|---|---|---|---|---|
| | | Test AUC | Applier (Exact) | Applier (RR) |
| Criteo (H='768,256,128', $B$=100k) | (False, False) | 0.79651 | **0.79711** | 0.79710 |
| | (False, True) | 0.79603 | 0.79728 | **0.79775** |
| | (True, False) | 0.79566 | 0.79755 | **0.79769** |
| | (True, True) | 0.79512 | 0.79693 | **0.79724** |
| Avazu (H='512,256,128', $B$=20k) | (False, False) | 0.77209 | 0.77348 | **0.77351** |
| | (False, True) | 0.77042 | 0.77217 | **0.77229** |
| | (True, False) | 0.77171 | 0.77358 | **0.77363** |
| | (True, True) | 0.77019 | **0.77235** | 0.77220 |
| MovieLens 1M (H='256,128', $B$=6k) | (False, False) | 0.84721 | 0.85246 | **0.85275** |
| | (False, True) | 0.82844 | 0.83343 | **0.83374** |
| | (True, False) | 0.84746 | **0.85310** | 0.85277 |
| | (True, True) | 0.82954 | 0.83348 | **0.83379** |

Table 2: Detailed ablation study (Test AUC) on all three datasets, using the same configurations as Table 1. We compare the final AUC of the Stage 1 **Search Model** against the two Stage 2 **Applier** variants (Exact Indices vs. RR Load-Balanced) trained from its discovered policy.

| Dataset | Config | Baselines (Recurring) | | Our Method (SEO) | |
|---|---|---|---|---|---|
| | | Sep. | MSE | Applier (RR) | Search (Stage 1) |
| Criteo | (H='768,256,128', B=100k) | 2,211.3s | **1,527.1s** | 1,738.3s | |
| | Search (DP=F, AH=F) | | | | 2,765.5s |
| | Search (DP=F, AH=T) | | | | 2,069.9s |
| | Search (DP=T, AH=F) | | | | 3,355.2s |
| | Search (DP=T, AH=T) | | | | 1,985.2s |
| Avazu | (H='512,256,128', B=20k) | 8,285.7s | **1,949.2s** | 2,122.1s | |
| | Search (DP=F, AH=F) | | | | 2,418.1s |
| | Search (DP=F, AH=T) | | | | 3,866.2s |
| | Search (DP=T, AH=F) | | | | 3,406.5s |
| | Search (DP=T, AH=T) | | | | 2,902.5s |
| MovieLens | (H='256,128', B=6k) | **99.3s** | 131.8s | 107.3s | |
| | Search (DP=F, AH=F) | | | | 97.2s |
| | Search (DP=F, AH=T) | | | | 173.6s |
| | Search (DP=T, AH=F) | | | | 109.0s |
| | Search (DP=T, AH=T) | | | | 142.2s |

Table 3: Training Time per Epoch (seconds) reported as the median over 5 runs. 'DP' refers to 'use_down_projection' and 'AH' to 'adjust_hash_buckets'. The fastest recurring cost (baselines vs. Applier (RR)) for each dataset is in **bold**.

the 'Separate' baseline. On MovieLens, our 'Applier' (107.3s) is faster than MSE (131.8s) and only marginally slower than the 'Separate' baseline (**99.3s**).

The 'Search' phase does introduce a one-time, offline computation cost. However, this cost is reasonable; for instance, on Criteo, the fastest search configuration (1,985.2s) is faster than the 'Separate' baseline. On both MovieLens and Criteo, the search cost is comparable to a single epoch of baseline training. This cost is incurred only once to determine the optimal embedding architecture and is amortized over all subsequent training runs of the more performant 'Applier' model. In summary, our two-stage approach's recurring training cost is highly competitive, demonstrating that the significant performance gains do not come with an unreasonable training overhead.

## 5 CONCLUSION

We proposed Shared Embedding Optimization (SEO), a two-stage (search-retrain) framework for learning optimal, heterogeneous embedding structures. Theoretically, its hypothesis space supersets the monolithic model, guaranteeing at least equal performance. Experimentally, our SEO Appliers consistently outperform 'Separate' and 'Monolithic Shared' baselines on Criteo, Avazu, and MovieLens 1M, given identical parameter budgets and comparable training times.

## ETHICS STATEMENT

Our research proposes a foundational method, Shared Embedding Optimization (SEO), for improving the parameter efficiency of embedding layers, primarily in deep learning recommender systems. We have evaluated our method using three well-established, anonymized, and publicly available benchmark datasets: Criteo, Avazu, and MovieLens 1M. Our study did not involve any new data collection or human subjects.

We acknowledge that recommender systems, as a class of applications, can have broader societal impacts, including potential fairness and bias issues. Our work focuses on the structural and computational efficiency of the embedding component, not the downstream recommendation logic itself. The learned embedding-sharing policy could, in principle, interact with model fairness, for example, by allocating representational capacity inequitably across different user groups. Our current study does not investigate the fairness or bias implications of the learned structures. We believe this is an important and necessary direction for future work, particularly analyzing how optimized embedding architectures like those found by SEO interact with fairness-aware training objectives and bias mitigation techniques. To our knowledge, our method does not introduce new privacy, security, or ethical risks beyond those already associated with standard deep learning models.

## REPRODUCIBILITY STATEMENT

We are committed to ensuring the reproducibility of our results. We will provide an anonymous link to our source code, models, and experimental scripts as supplementary material. This repository will contain implementations for our two-stage SEO framework's Applier stage (Stage 2), all baseline models (Separate and Monolithic Shared), and the experimental harnesses. The Search stage (Stage 1), as described in Section 3, utilizes a differentiable search mechanism adapted from Yasuda et al. (2024). While our specific internal implementation of this search component cannot be open-sourced due to proprietary constraints, we will provide detailed pseudo-code and a thorough description in the appendix to facilitate reimplementation. Furthermore, we will release the exact architectural policies discovered by our search for each experiment, allowing the community to directly run and verify the Applier model, which is trained from scratch and represents a core component of our contribution.

Our experiments are conducted on three public benchmarks (Criteo, Avazu, MovieLens 1M). We detail our preprocessing and data splitting protocols in Section 4.1, which follow established prior work (Wang et al., 2021; Song et al., 2019) to facilitate direct comparison. The theoretical justification for our framework, introduced in Section 3.3, is accompanied by detailed proofs in Appendix C. All critical hyperparameters, such as MLP architecture, base bucket size, and optimizer settings, are described in Section 4.1. Furthermore, we provide a comprehensive set of results, including detailed ablation studies (Table 2) and the full experimental outcomes across all tested hyperparameter configurations in Appendix D, ensuring transparency and allowing for thorough verification of our claims.

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

## A  STATEMENT ON LLM USAGE

In accordance with the ICLR 2026 policy, we report that Large Language Models (LLMs) were used as a general-purpose writing assistant for this paper. The use of LLMs was strictly limited to aiding with and polishing the writing, such as improving grammar, clarity, and conciseness. LLMs did not contribute to the research ideation, methodology, experimental analysis, or the generation of core results. All authors have reviewed and take full responsibility for the final content.

## B  FURTHER RELATED WORK

This section provides additional context on related research areas that inform our work but are less directly comparable to the specific problem solved by SEO.

### B.1  FOUNDATIONAL EMBEDDING COMPRESSION

Foundational techniques aim to reduce the size of a pre-trained model, typically by lowering the numerical precision of its parameters or removing them entirely. The seminal work of Han et al. (2016) introduced a three-stage pipeline of pruning, trained quantization, and Huffman coding, demonstrating significant model size reduction on computer vision tasks without accuracy loss. This established a powerful post-hoc compression paradigm. These methods are fundamentally different from our Shared Embedding Optimization (SEO) framework. Pruning and quantization are *post-hoc* optimizations applied to an already trained model; their goal is to minimize model size subject to a constraint on accuracy degradation. In contrast, SEO is a *pre-training structural optimization*. It operates under a strict constraint of a **fixed parameter budget** with the goal of *improving* model accuracy.

### B.2  HASHING VARIANTS AND STRUCTURED SHARING

The hashing trick is a cornerstone of parameter-efficient embeddings, but its random nature has led to the development of more sophisticated variants. *Hash embeddings* (Svenstrup et al., 2017) assign each feature to multiple rows in the table, with the final embedding being a weighted sum. Other methods modify the lookup process itself. For instance, *HashedNet* (Chen et al., 2015) independently looks up each dimension of the embedding in a flattened parameter space, while *deep hash embeddings* (Kang et al., 2021) use a neural network to directly output the embedding vector. Beyond hashing, other structured approaches exist. Compositional embeddings construct a unique representation for each feature by combining vectors from multiple smaller, shared codebooks, often using a fixed rule like summation or element-wise products (Shi et al., 2020). Another class of methods uses tensor decomposition to re-parameterize a large embedding table as a product of smaller, low-rank tensors. While SEO is conceptually related, with our "chunks" being analogous to shared codebooks, a key distinction is that these methods typically employ a *fixed, pre-defined* combination rule. SEO, in contrast, uses its Stage 1 Search phase to *learn* an optimal, feature-specific combination policy.

## C  PROOFS

*Proof of Theorem 1.* We prove $\mathcal{M}_H \subseteq \mathcal{M}_A$ by construction. Let $M_H$ be an arbitrary model in the MSE hypothesis space $\mathcal{M}_H$, defined by its parameters $\boldsymbol{\theta}_H = (W_H, W_{MLP})$, where $W_H \in \mathbb{R}^{B \times D}$. We construct a specific model $M_A \in \mathcal{M}_A$ as follows:

1. Set the Applier's MLP parameters to be identical to the MSE model's parameters: $W'_{MLP} = W_{MLP}$.

2. Logically partition the MSE embedding table $W_H$ into $M$ chunks $T_1, \ldots, T_M$ such that $W_H = \text{Concat}(T_1, ..., T_M)$. We set the Applier's embedding table $W_A = W_H$.

3. Define the structural policy $\mathcal{S}$ for $M_A$ as the "trivial policy" $\mathcal{S}_{trivial}$, where for every feature $f \in \mathcal{F}$, the policy is to select all chunks: $\mathcal{S}_f = \{1, ..., M\}$.

This constructed model $M_A$ is, by definition, a member of the Applier hypothesis space $\mathcal{M}_A$. We verify that it computes an identical function to $M_H$. For any feature $f$, the embedding function of $M_H$ is $E_f(\mathbf{x}) = \text{Lookup}(W_H, \mathbf{x}_f) = \text{Concat}(T_1(\mathbf{x}_f), ..., T_M(\mathbf{x}_f))$. The embedding function of our $M_A$ is $E_f(\mathbf{x}, \mathcal{S}_{trivial}) = \text{Concat}(\{T_i(\mathbf{x}_f) \mid i \in \mathcal{S}_{trivial,f}\}) = \text{Concat}(T_1(\mathbf{x}_f), ..., T_M(\mathbf{x}_f))$. Since the embedding outputs are identical for all features and the MLP weights are identical, the models compute the same function for all inputs. Thus, any $M_H \in \mathcal{M}_H$ has an equivalent $M_A \in \mathcal{M}_A$. The inclusion is strict ($\mathcal{M}_H \neq \mathcal{M}_A$) because $\mathcal{M}_A$ also contains all heterogeneous models where $\mathcal{S}_f \neq \{1, ..., M\}$ for at least one feature $f$ (e.g., $\mathcal{S}_f = \{1\}$). These models are not representable by any model in $\mathcal{M}_H$. □

*Proof of Corollary 1.* This follows directly from Theorem 1. Since $\mathcal{M}_H \subset \mathcal{M}_A$, the minimization over the superset $\mathcal{M}_A$ must find a solution at least as good as (and potentially better than) the minimization over the subset $\mathcal{M}_H$. □

## D FULL HYPERPARAMETER RESULTS (TEST AUC)

This appendix contains the full Test AUC comparison plots for all 'Base Bucket Size' ($B_{base}$) configurations tested for each dataset, complementing the representative results shown in Figure 2.

### D.1 MOVIELENS 1M RESULTS

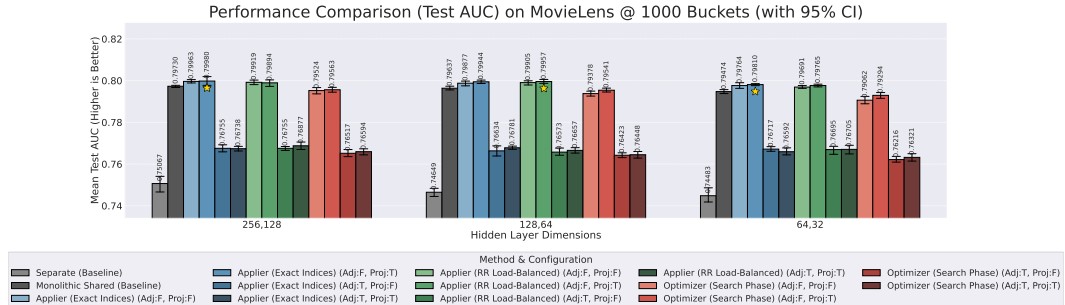

Figure 3: MovieLens (Test AUC) @ 1k Base Bucket Size

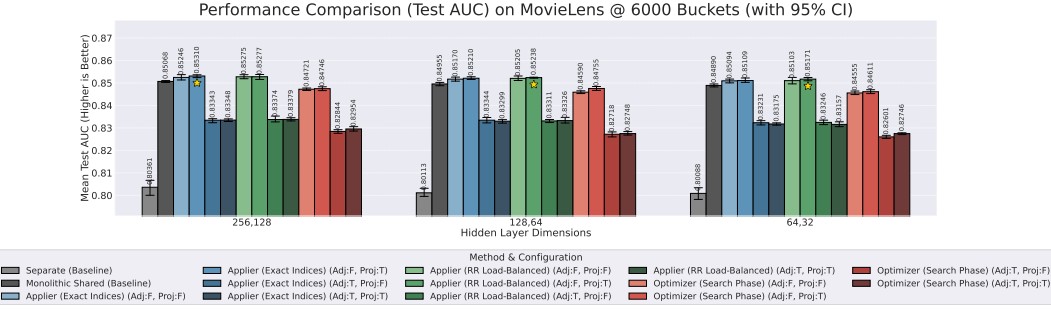

Figure 4: MovieLens (Test AUC) @ 6k Base Bucket Size (Same as Figure 2a)

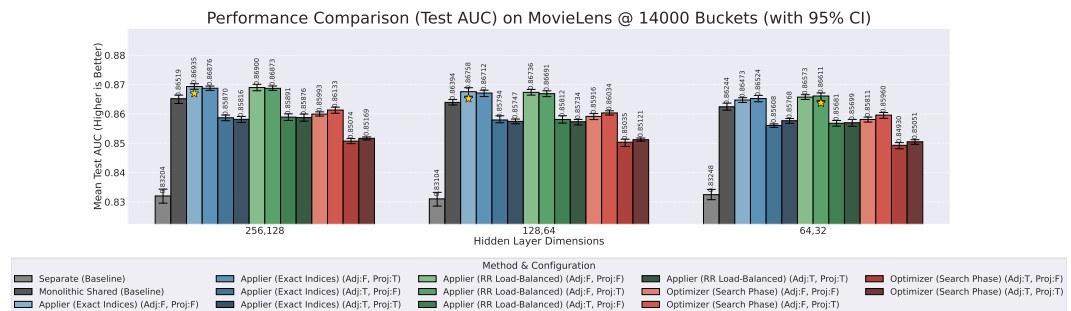

Figure 5: MovieLens (Test AUC) @ 14k Base Bucket Size

## D.2 AVAZU RESULTS

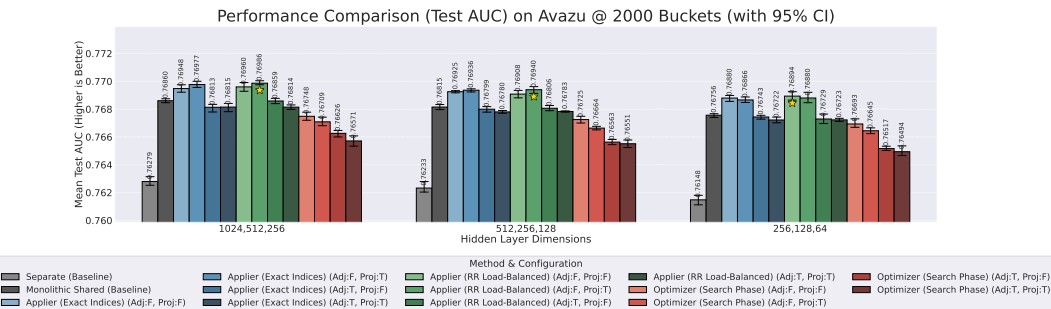

Figure 6: Avazu (Test AUC) @ 2k Base Bucket Size

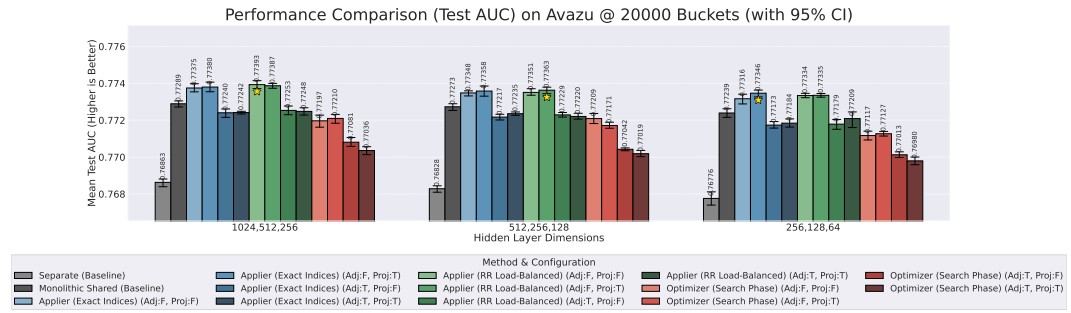

Figure 7: Avazu (Test AUC) @ 20k Base Bucket Size (Same as Figure 2b)

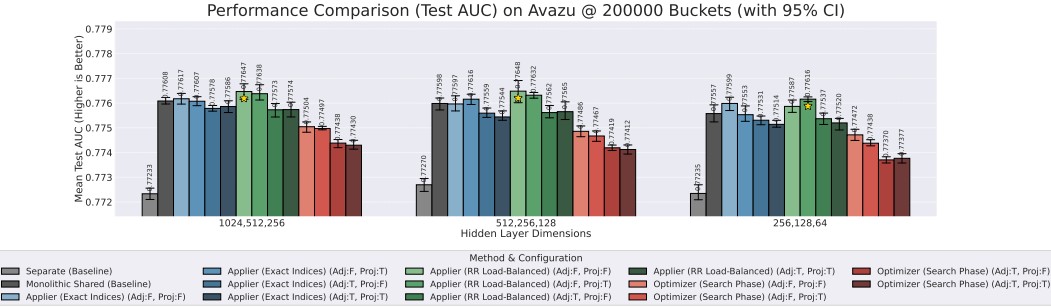

Figure 8: Avazu (Test AUC) @ 200k Base Bucket Size

## D.3 CRITEO SMALL RESULTS

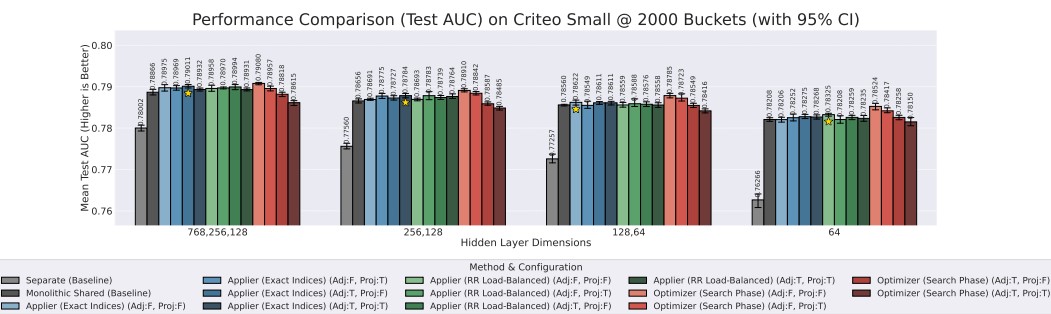

Figure 9: Criteo Small (Test AUC) @ 2k Base Bucket Size

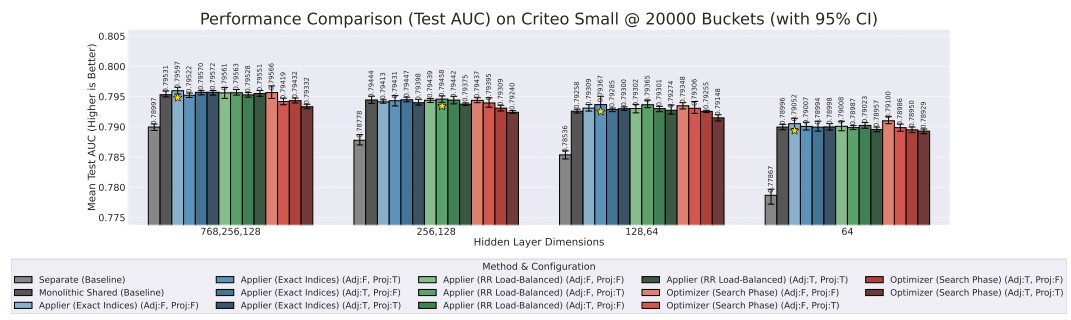

Figure 10: Criteo Small (Test AUC) @ 20k Base Bucket Size

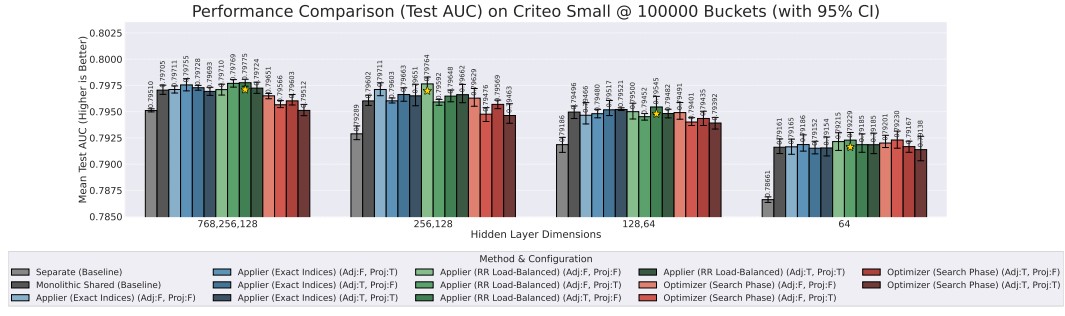

Figure 11: Criteo Small (Test AUC) @ 100k Base Bucket Size (Same as Figure 2c)

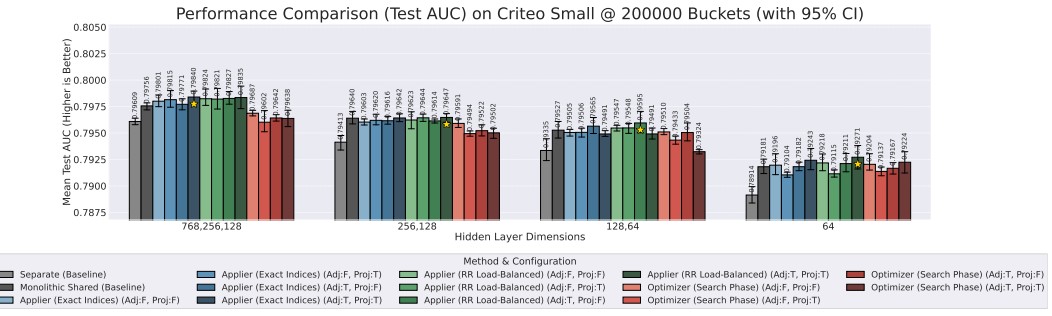

Figure 12: Criteo Small (Test AUC) @ 200k Base Bucket Size

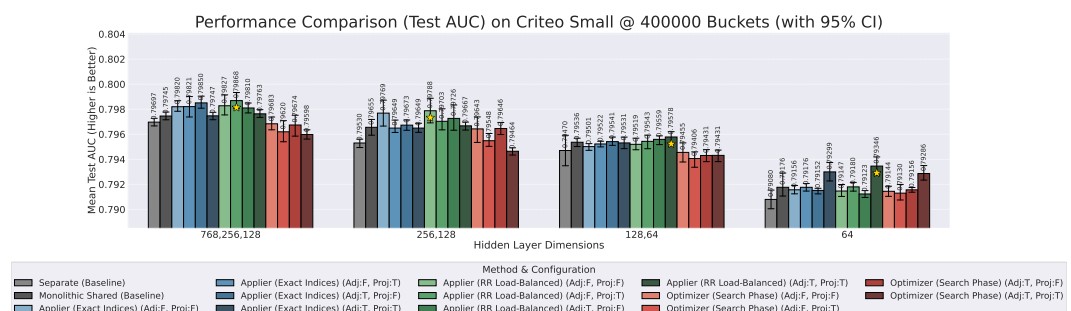

Figure 13: Criteo Small (Test AUC) @ 400k Base Bucket Size

