# OpenReview forum: "Shared Embedding Optimization: A Two-Stage Approach for Efficient and Effective Feature Embedding"
_ICLR.cc/2026/Conference — ICLR 2026 Conference Withdrawn Submission_

### Official Review · Reviewer_6PDq · 2025-10-30

**Soundness:** 2
**Presentation:** 3
**Contribution:** 2
**Rating:** 4
**Confidence:** 4

**Summary:**

The paper presents a two-stage search and retrain framework to improve embedding learning in large-scale recommendation systems. Specifically, the authors propose SEO, which can be considered a heterogeneous configuration between the Separate and Monolithic Shared frameworks. It consists of a search stage to learn feature-specific policies for allocating embedding chunks, and a applier stage to re-train the model under the same parameter budget. Experiments in several public datasets demonstrate the effectiveness of the proposed approach.

**Strengths:**

1. Motivation. The paper tackles a core challenge in large-scale recommendation systems: the trade-off between parameter efficiency and the effectiveness of the embedding learning.

2. Theoretical Justification. The paper proves that SEO’s Applier space is a strict superset of monolithic embeddings, giving a theoretical guarantee that it can achieve at least equal or better performance.

3. Presentation. The whole paper is organized well, with clear presentation of the motivation, and detailed elaboration of the propposed method.

**Weaknesses:**

1. Experiments. The evaluation is based on the DNN model, which is not strong enough. Experiments on recent SOTAs like DCN V2, xDeepFM, Wukong, and so on would be great.

2. Down Projection. With the Down Projection, the rank of the original long embeddings is reduced, which makes learning in the large dimension space unnecessary. Besides, this projection operator may introduce extra computational cost.

3. Technical Contribution. The novelty of the proposed method, i.e., pruning the embedding in a two-stage framework, is limited.

4. Analysis. Why is the learned embedding by the proposed approach better than existing works? In-depth analysis on the quality of the embeddings from representation perspectives (such as Information Bottleneck [1] or Dimensionally Robustness [2,3] ) would be great.

[1]. Deep Learning and the Information Bottleneck Principle.

[2]. On the embedding collapse when scaling up recommendation models. ICML 2024.

[3]. Towards Mitigating Dimensional Collapse of Representations in Collaborative Filtering. WSDM 2024.

**Questions:**

see the weakness

---

### Official Review · Reviewer_UAzf · 2025-10-31

**Soundness:** 2
**Presentation:** 3
**Contribution:** 2
**Rating:** 2
**Confidence:** 4

**Summary:**

This paper aims to resolve the "performance-efficiency dilemma" for embedding tables in large-scale recommendation systems. The authors argue that practitioners are currently forced to compromise between two extremes: "Separate" embeddings (good performance, low parameter efficiency) and "Monolithic Shared" embeddings (high parameter efficiency, poor performance). The paper contends this is a "false dilemma" and that the optimal solution is a heterogeneous sharing structure. To this end, the authors propose "Shared Embedding Optimization" (SEO), a two-stage framework. Stage 1 (Search) uses a differentiable search on an over-parameterized model to learn an optimal, feature-specific policy for allocating and sharing embedding "chunks". Stage 2 (Applier) discards the search model and uses the learned policy blueprint to instantiate a new, compact model, which is then retrained from scratch. Experiments on three large-scale benchmarks (Criteo, Avazu, MovieLens 1M) demonstrate that under an **identical parameter budget** and **comparable recurring training time**, the SEO framework significantly and consistently outperforms both the "Separate" and "Monolithic Shared" baselines.

**Strengths:**

1. **Novel and Practical Two-Stage Framework:** The paper proposes a clear and decoupled "Search-then-Retrain" framework (SEO). This approach separates the complex task of structural discovery (Stage 1) from the final weight optimization (Stage 2) to automatically learn a task-specific, heterogeneous embedding sharing structure. This provides a very practical new direction for resolving the long-standing "Separate vs. Monolithic" dilemma.

2. **Rigorous and Strong Empirical Results:** The method is validated on three large-scale public benchmarks (Criteo, Avazu, MovieLens 1M). The experimental results (e.g., Table 1, Figure 2) consistently show that under a **strictly controlled identical parameter budget**, the final SEO Applier model significantly outperforms both "Separate" and "Monolithic Shared" baselines on both AUC and LogLoss. This strongly validates the method's effectiveness in improving performance-per-parameter.

**Weaknesses:**

**1. Baseline Insufficiency and "Setting Avoidance":** The most critical weakness of this study is its insufficient baseline comparison, which suggests a "setting avoidance" problem. The authors correctly identify the "Separate" vs. "Monolithic Shared" (MSE) paradigm as a "false dilemma" in the introduction, yet they **only** compare SEO against these two extreme baselines in the experiments. By strictly limiting the comparison to a "single-table + equal-parameter" setting, the authors cleverly avoid head-to-head comparisons with the SOTA parameter-efficient embedding methods that are truly relevant to both industry and academia. A method claiming to optimize shared structures must be empirically compared against strong baselines, such as: (1) industrial single-table solutions (e.g., a proxy for Unified Embedding), (2) other AutoML embedding methods (e.g., AutoDim / AutoEmb), (3) compositional/hashing variants (e.g., Compositional Embeddings, Codebook, or HashedNet), and (4) pruning-derived baselines (e.g., an MSE model pruned and fine-tuned to the same parameter count, and unstructured pruning methods such as PEP). Without these comparisons, SEO's victory is predictable but fails to prove its superiority over established strong baselines, giving the impression of "winning by convenient problem definition."

**2. Misleading Total Cost of Ownership (TCO) Analysis:** The analysis of the Total Cost of Ownership (TCO) is misleading. The paper (Table 3) strategically focuses its cost analysis on the "recurring training cost" (Stage 2 Applier), but this ignores the **one-time, yet potentially exorbitant, search cost** of Stage 1. For instance, the paper claims the search cost is "reasonable" but only reports the "per-epoch" time for the search. It **fails to report how many epochs Stage 1 requires to converge** to a good policy. If the search phase requires 100 epochs, this "one-time cost" could be 100x that of a single recurring run, making the method impractical for many applications. This selective reporting obscures the true TCO and makes fair cost comparison against methods like post-hoc compression impossible.

**3. Lack of Diagnostic and Ablation Studies:** The paper provides insufficient diagnostic analysis to disentangle the sources of its performance gains or test its robustness. The SEO framework introduces new "meta-hyperparameters" that are not ablated, such as the search dimension ($D_{search}$), chunk size ($d_{chunk}$), and the search mechanism's own hyperparameters (e.g., temperature $\tau$). More importantly, the paper lacks key "source of gain" ablations, for example: (1) A "Random-chunk" baseline that preserves the "capacity quota" (chunks-per-feature) learned by SEO but assigns the specific chunk indices randomly; this would separate the contribution of "quota learning" from "specific structure learning." (2) A frequency/long-tail analysis of SEO's performance on low-frequency (cold-start) vs. high-frequency features. (3) Diagnostic visualizations such as hash collision rates or chunk utilization heatmaps. Without these analyses, it is difficult to confirm that the gains come from "structural learning" itself rather than from a "fortuitous hyperparameter setup."

**4. Weak Theoretical Contribution:** The paper's theoretical contribution is weak and lacks explanatory power. The paper highlights its "theoretical justification" (Section 3.3) as a contribution, which proves that the Applier's hypothesis space $M_A$ is a superset of the Monolithic space $M_H$. However, as the proof in Appendix C shows, this theorem is **"trivial"**, as it merely states that the Applier *can* simulate the Monolithic model by selecting all chunks. This theory **provides no guarantees about the effectiveness of the search process itself**. More importantly, it lacks a more macroscopic explanation for *why* the learned heterogeneous structure should practically outperform the homogeneous one after SGD training. The paper should attempt to supplement this with analysis from the perspective of capacity allocation, collision probability, or frequency-weighted capacity matching.

**Questions:**

None

---

### Official Review · Reviewer_K1Q3 · 2025-11-01

**Soundness:** 3
**Presentation:** 3
**Contribution:** 3
**Rating:** 8
**Confidence:** 3

**Summary:**

Authors propose Shared Embedding Optimization (SEO), a two-stage framework for learning heterogeneous structures within a unified embedding table for categorical features in Recsys models. Stage 1 - differentiable search over an over-parameterized table to derive feature-specific chunk allocation policies, then Stage 2 - retrain a compact model from scratch using exact or load-balanced variants.

The authors provide a theoretical justification that the Applier's hypothesis space is a strict superset of the monolithic model's, guaranteeing a solution of at least equal quality. Comprehensive experiments on three public benchmarks (Criteo, Avazu, MovieLens 1M) demonstrate that the SEO Applier models consistently and significantly outperform both the "Separate" and "Monolithic Shared" baselines

**Strengths:**

The work targets a practical bottleneck in recommender systems: the parameter inefficiency of embedding tables, which can dominate model size. The authors creatively adapted differentiable structured pruning techniques to embedding architecture search, enabling heterogeneous sharing that mitigates the "one-size-fits-all" limitations of monolithic embeddings.

The theoretical justification—that the Applier's hypothesis space is a superset of monolithic baselines—provides a clear optimization guarantee, and the two-stage decoupling of search from retraining is a sensible design choice. Experimentally, the setup is fair with fixed budgets, and ablations.

Insightful Ablations: The ablation study in Table 2 is strong, not only by proving the two-stage process is necessary, but also by revealing a nuanced and interesting finding like the Applier (RR Load-Balanced) variant is best for large, sparse datasets (Criteo, Avazu), while Applier (Exact Indices) is best for the smaller, denser dataset (MovieLens).

**Weaknesses:**

Could be strengthened by comparing with work like fine-grained dimension pruning in FIITED or regularization in Stochastic Shared Embeddings (NeurIPS 2019), which achieve similar goals of parameter-efficient representations.

Scalability and Cost of Stage 1: The proposal analyzes the recurring cost of Applier model but is less detailed about the scalability of the one-time Stage 1 search. The search space size appears to be a function of the number of features and candidate chunks. It is unclear how this search cost would scale to a real-world system with thousands of features.

**Questions:**

Search Space Hyperparameters: The search is performed in an "over-parameterized" space, for which the authors chose D=32 (double the baseline D=16). This choice seems somewhat arbitrary. There is no ablation on how this D affects the quality of the final policy found. Would a wider search (D=64) find a better, more robust policy? Or is this expansion even necessary?

Applier Variant Choice: The finding that (RR) is better for some datasets and (Exact) is better for others is a good observation, but it remains a post-hoc one. Do the authors have any "rules of thumb" or dataset characteristics (feature cardinality distributions, sparsity) that could help a practitioner predict which Applier variant (Exact vs. RR) is likely to perform better on a new, unseen dataset?

---

### Official Review · Reviewer_mhB6 · 2025-11-01

**Soundness:** 2
**Presentation:** 2
**Contribution:** 2
**Rating:** 2
**Confidence:** 3

**Summary:**

This paper addresses the memory and computational bottlenecks of embedding tables in large-scale recommendation systems by proposing a novel two-stage framework called Shared Embedding Optimization (SEO). The method aims to move beyond the conventional trade-off between "fully separated" and "fully shared" strategies, seeking to enhance model performance under a fixed parameter budget， heterogeneous embedding sharing structure. The paper provides theoretical justification for the framework and validates its effectiveness through experiments on relevant benchmark datasets.

**Strengths:**

1. Novelty: The proposed Shared Embedding Optimization (SEO) framework offers a fresh approach to addressing the memory and computational bottlenecks of embedding tables, breaking through the limitations of traditional "fully separate" and "fully shared" strategies by automatically learning a heterogeneous sharing structure.

2. Theoretical Foundation: The paper provides theoretical justification for the proposed method, thereby offering solid theoretical support.

**Weaknesses:**

1. Limited Significance of Experimental Results: The performance improvements reported in the paper are marginal, and in some cases, the method fails to outperform baseline models. This raises concerns about the method's effectiveness and stability—whether the observed minor improvements are statistically significant or merely due to random fluctuations.

2. Insufficient Baseline Comparisons: The baseline methods selected in the experimental design are mostly traditional and basic models, lacking systematic comparison with more advanced and competitive approaches in the field.

3. Questions about Methodological Consistency: The "Choice of Applier Variant Matters"  proposed in the paper appears closer to a summary of experimental observations across different datasets rather than the construction of a universal principle. It remains unclear whether its core mechanism truly achieves automated decision-making or still relies on exhaustive trial-and-error for new datasets.

4. Formatting Issues: The presence of Chinese characters in certain parts of the manuscript requires careful review and correction to adhere to formatting standards.

**Questions:**

1.  The performance improvements reported in the paper are quite marginal, and in some experimental settings the method fails to outperform the baseline models. Could you provide more convincing evidence to demonstrate the reliability of these performance differences? How do you explain the instability of the method's performance across different datasets?

2.  The baseline methods selected for the experiments are mostly traditional and fundamental models, lacking systematic comparison with more advanced embedding optimization approaches in the field. Could you supplement the experiments with comparisons against at  recent representative methods?

3.  The argument that the "choice of applier variant matters" appears more like a summary of experimental phenomena across different datasets rather than a generalizable principle. Could you clarify whether your method requires re-establishing decision principles when facing a completely new dataset?

4.  Could you analyze the specific reasons for the performance degradation observed on some datasets? Is this related to specific dataset characteristics or hyperparameter choices? Does the method have clear applicability boundaries?

5.  The framework provides an extensive set of configurable options. Have you considered simplifying these by identifying a core subset that maintains effectiveness while reducing complexity?

---

### Note · Authors · 2026-01-02

I have read and agree with the venue's withdrawal policy on behalf of myself and my co-authors.